# Jak2 and Jaw Muscles Are Required for Buccopharyngeal Membrane Perforation during Mouth Development

**DOI:** 10.3390/jdb11020024

**Published:** 2023-05-31

**Authors:** Amanda J. G. Dickinson

**Affiliations:** Department of Biology, Virginia Commonwealth University, Richmond, VA 23284, USA; ajdickinson@vcu.edu

**Keywords:** buccopharyngeal membrane, choanal atresia, mouth, *Xenopus laevis*

## Abstract

The mouth is a central feature of our face, without which we could not eat, breathe, or communicate. A critical and early event in mouth formation is the creation of a “hole” which connects the digestive system and the external environment. This hole, which has also been called the primary or embryonic mouth in vertebrates, is initially covered by a 1–2 cell layer thick structure called the buccopharyngeal membrane. When the buccopharyngeal membrane does not rupture, it impairs early mouth functions and may also lead to further craniofacial malformations. Using a chemical screen in an animal model (*Xenopus laevis*) and genetic data from humans, we determined that Janus kinase 2 (Jak2) has a role in buccopharyngeal membrane rupture. We have determined that decreased Jak2 function, using antisense morpholinos or a pharmacological antagonist, caused a persistent buccopharyngeal membrane as well as the loss of jaw muscles. Surprisingly, we observed that the jaw muscle compartments were connected to the oral epithelium that is continuous with the buccopharyngeal membrane. Severing such connections resulted in buccopharyngeal membrane buckling and persistence. We also noted puncta accumulation of F-actin, an indicator of tension, in the buccopharyngeal membrane during perforation. Taken together, the data has led us to a hypothesis that muscles are required to exert tension across the buccopharyngeal membrane, and such tension is necessary for its perforation.

## 1. Introduction

The mouth is a central feature of our face, facilitating our most important functions such as eating, breathing, and speech. An important event in mouth formation is the creation of a “hole” which connects the digestive system and the external environment. This hole, which has also been called the primary or embryonic mouth in vertebrates [1,2,3], appears in humans during the fourth week of pregnancy. At this time, the surrounding face is still in the early stages of development, comprising the frontonasal and paired maxillary and mandibular prominences. These prominences will grow, converge, and develop into the tissues such as bones and muscles that make up the jaw and palate (see reviews [4,5,6,7,8]). As the craniofacial complex matures, the remnants of the embryonic mouth will then reside deep inside the oral cavity at the level of the tonsils. Due to the transient nature and the difficulty of studying the embryonic mouth in mammals, the importance of the structure has been underappreciated. However, a revival of embryological and molecular studies focusing on the formation of the mouth in other developmental models has revealed its importance in craniofacial development. For example, the region that gives rise to the embryonic mouth, called the Extreme Anterior Domain, has been shown to contain signals that pattern the surrounding face in *Xenopus* [9,10,11]. On the other hand, important aspects of the mouth opening have been proposed but little studied. For example, the actual hole formed during embryonic mouth formation could provide the physical separation of the growing prominences, yet this has not been rigorously tested. In mammals, the opening to the digestive system is also thought to be required for ingestion of amniotic fluid which aids in the development of both the digestive and immune systems [12]. However, this role is very difficult to study. Thus, it is clear we need more comprehensive studies of the anatomy and formation of the mouth.

*Xenopus* has been integral to furthering our basic understanding of how the mouth is formed. In this species, the mouth develops similarly to reptiles, birds, and mammals [3]. Xenopus has been instrumental in identifying the Extreme Anterior Domain, a specialized signaling center that gives rise to the mouth [9,10,11]. Furthermore, in *Xenopus*, we have also defined several morphological changes or processes that lead to mouth formation [1,2,9,13,14,15]. Such processes include cell death and cellular reorganizations that eventually form a thin covering over the embryonic mouth called the buccopharyngeal membrane. This buccopharyngeal membrane, a structure observed across vertebrates, is actually 1–2 cell layers which are continuous with the facial epidermis and lining of the oral cavity [3,16,17,18,19,20]. The cells of the buccopharyngeal membrane are unique in that they do not secrete an ECM and are not derived from mesodermal nor neural crest cells [9]. An important last step in embryonic mouth development is the rupture or perforation of this buccopharyngeal membrane which then creates the opening to the digestive system.

When the buccopharyngeal membrane does not rupture, it results in a medical condition called persistent buccopharyngeal membrane [21,22,23,24]. While rare on its own, this condition also appears to be present in a number of craniofacial syndromes such as 1p36 deletion syndrome (OMIM #607872), hypomandibular faciocranial dysostosis (OMIM #241310) agnathia–otocephaly complex (OMIM #202650), microphthalmia, syndromic 5 (OMIM # 610125), and Holzgreve–Wagner–Rehder syndrome (OMIM # 236110) [25,26]. A persistent buccopharyngeal membrane can also be accompanied by a cleft palate [27]. Importantly, failure of the buccopharyngeal membrane to rupture has also been proposed to be a possible cause for more common oral obstruction-type malformations [28,29,30]. In particular, one such condition, Choanal Atresia, occurs when there is a physical blockage between nasal and oral cavities at birth [31,32,33]. This condition can cause respiratory distress in newborns and when undiagnosed has been postulated to lead to sudden infant death syndrome (SIDS) [34]. Choanal Atresia is difficult to correct and even with surgery, patients have long-lasting problems in their airways that affect their quality of life [31]. Strikingly, in a search of the OMIM database, Choanal Atresia is present in 82 different congenital disorders (Appendix A). Human and mouse studies have revealed that disruptions of retinoic acid and FGF signaling can result in Choanal Atresia [32,33,35]. Such studies also indicate that disruptions to other craniofacial processes such as primitive choana formation and cranial vault development can lead to this craniofacial malformation. Thus, if a persistent buccopharyngeal membrane is responsible for even a subset of the multitude of disorders accompanied by Choanal Atresia, there is an even more compelling reason to understand mechanisms that affect the last stages of mouth development.

In summary, a collation of case studies and descriptions of craniofacial disorders has revealed that a persistent buccopharyngeal membrane may be more pervasive and detrimental than initially believed. However, we know very little about the mechanisms governing the process that leads to the rupture of this structure. In fact, there has only been a handful of studies in the last 20 years devoted to mouth development [2,3,9,11,12,13,14,15,36,37,38]. The goals of this study are to begin to rectify a gap in our knowledge of mouth development and uncover potential mechanisms that drive the perforation of the buccopharyngeal membrane. We use the vertebrate model *Xenopus laevis*, an ideal model for studies of buccopharyngeal membrane rupture. Not only is this the best-studied vertebrate in terms of mouth development, development of the mouth is also very similar to mammals [3]. Moreover, the mouth is easier to view and study in *Xenopus* embryos compared to other developmental models. We use comprehensive chemical and genetic screens combined with classic embryological methods to demonstrate that rupture of the buccopharyngeal membrane requires mechanical forces generated by the jaw muscles.

## 2. Materials and Methods

### 2.1. Gathering Embryos and Staging

*Xenopus laevis* were housed in the Biology Department Aquatics facility and maintained using protocols approved by VCU IACUC. Embryos were obtained and cultured using standard methods [39]. To ensure that results were not clutch-specific, we gathered eggs from two to three females into a petri dish of egg media (high salt 1× Modified Barth’s Solution). The media was removed and one half of a testis (previously removed from a male frog) was cut up and spread around the dish of eggs. This was followed by adding frog embryo media (0.1× Modified Barth’s Solution). Once the embryos were fertilized, they were immersed in a 2% cysteine solution (pH 7.8) and swirled until the jelly coats were dissolved. Embryos were then washed in frog media and maintained in this solution in a 15 °C incubator until the desired stage for experiments. Staging was performed according to Nieuwkoop and Faber [40,41].

### 2.2. Chemical Screen

A chemical screen was performed using the Cancer Diversity Set II from the NCI Development Therapeutics Program. All chemicals are listed in the Appendix A and more information can be found at (https://dtp.cancer.gov/organization/dscb/obtaining/available_plates.htm, accessed on 1 May 2023). At stages 22–24, six embryos were placed in 1 mL of frog embryo media in each well of 48-well culture dishes. Stock solutions (10 mM) dissolved in DMSO were added directly to the wells so that the final concentrations of all chemicals were 30 μM. The culture dishes were gently swirled and then the dish was placed in an incubator set to 23 °C. Controls consisted of sibling embryos that were incubated with an equivalent volume (3 μL) of DMSO in the same volume of media. At stages 40–41, the media and chemicals were removed and replaced with 4% paraformaldehyde (PFA) to fix the embryos. The faces of embryos were imaged as described in Section 2.4, Imaging the Facial Features. Characteristics that were recorded were whether the embryos had (1) stomodeum, (2) an open mouth, (3) a cleft-like appearance of the open mouth, or (4) no observable sign of a mouth. All chemicals that caused embryos to have a stomodeum and persistent buccopharyngeal membrane were retested using the same protocol. Each chemical used in the screen has been assigned an NSC (National Service Center) number by the Development Therapeutics Program. The NSC numbers of the chemicals of interest were entered into the PubChem (https://pubchem.ncbi.nlm.nih.gov, accessed on 15 September 2016) database. This allowed us to identify candidate targets of chemicals that caused buccopharyngeal membrane persistence. Genecards (www.genecards.org, accessed on 15 September 2020 to 1 December 2020) were used to determine the function and corresponding gene symbol of each target. Functional classification of all the targets was performed using PANTHER (www.pantherdb.org, accessed on 23 May 2023) [42]. These data are available in the Appendix A.

### 2.3. Copy Number Variant Screen in Patients with Choanal Atresia

To collate all genes affected by copy number variations in patients with Choanal Atresia, we used the DECIPHER database (https://www.deciphergenomics.org, accessed on 1 May 2021 [43]. Genes were sorted into those represented in the most patients. Duplicates were also removed, and a functional analysis was performed using PANTHER. Data are available in the Appendix A.

### 2.4. Imaging the Facial Features

At stages 40–41, embryos were anesthetized in 1% tricaine for 10 min and then fixed in 4% paraformaldehyde overnight at 4 °C. Embryos were washed in phosphate-buffered saline with 0.1% Tween 20 (PBT) and placed in plastic petri dishes. A sterile disposable No. 15 scalpel (VWR, Radnor, PA, USA, cat. no: 82029-856) and Dumont No. 5 forceps (Fisher Scientific, Hampton, NH, USA, cat. no: NC9404145) were used to make two cuts to isolate the head: the first at the posterior end of the gut and then the second caudal to the gills. Isolated heads were mounted in small holes or depressions in either agarose- or clay-lined dishes containing PBT. The faces were imaged using a Discovery V8 stereoscope fitted with an Axiovision digital camera (Zeiss, Oberkochen, Germany).

### 2.5. Morpholino

Antisense *jak2* morpholino (*jak2* MO;CTCTCTGTTTGTGTCACTGTCAAAG), *myod* morpholino (*myod* MO; GGCAAGAGCTCCATAGAAACAGCCG [44] and standard control morpholino were purchased from Gene Tools (Philomath, OR, USA). Microinjections were carried out using an Eppendorf (Enfield, MA, USA) microinjector and a Zeiss stereoscope. 

### 2.6. Phalloidin Labeling, Immunofluorescence Auto Fluorescent Rendering

To examine F-actin, embryos were fixed in 4% PFA, washed in PBT, and incubated in Alexa Fluor 488 labeled Phalloidin (1:50, ThermoFisher, Waltham, MA, USA,) for 2–5 days at 4 °C. For immunofluorescence, embryos were fixed in 4% PFA then dehydrated in a methanol series (50%, 70%, 80%, 90%, and 100%) where they were then stored in the freezer (−20 °C) for 1 day to 2 weeks. Embryos were then rehydrated by reversing the same series of methanol solutions and then either processed whole or in tissue sections (see Section 2.7). The embryos or tissues were washed in PBT and blocked/permeabilized overnight in 1% BSA, 1% serum, and 1% triton-X 100. Samples were incubated in phospho-JAK2 (Millipore-Sigma, Burlington, MA, USA, 1:100), laminin antibody (Millipore-Sigma, Burlington, MA, USA, 1:1000), 12/101 (1:25, Developmental Studies Hybridoma Bank, Iowa city, IA, USA), or cleaved caspase-3 (Cell Signaling, Danvers, MA, USA, 1:1000) for 48 h and then washed in PBT. Secondary detection was performed with goat-anti-mouse or goat-anti-rabbit Alexa Fluor 488 or 568 (1:500, Invitrogen, Waltham, MA, USA) and in some cases combined with 2 drops of Dapi (ThermoFisher, Waltham, MA, USA) for 24 h. After washing in PBT, samples were immersed in 90% glycerol in PBT for imaging.

Embryos were rendered auto-fluorescent by immersing in Bouin’s fixative for one day to two weeks. After washing in 100% ethanol, embryos were then immersed in a solution of two parts benzyl benzoate and one part benzyl alcohol and imaged on glass slides.

All fluorescent images were collected with a C2 Nikon confocal microscope (VCU Biology microscopy core, Richmond, VA, USA) using 0.5–1.0 micron steps and compiled using the maximum intensity function to compress the z stacks. All experimental images were adjusted for brightness and color balance in the same way as controls.

### 2.7. Tissue Sectioning

Embryos were sectioned using two different methods (1) thick agarose sections and (2) thin plastic sections. Thick agarose sections were created with embryos fixed in 4% PFA and then washed in PBT. Embryos were then immersed in 4–5% low-melt agarose in a small petri dish. A square of agarose was cut, containing the embryos, and attached using superglue to the mounting block. Then, 200-micron sections were created with a 5000 series Leica vibratome. For thin plastic sections, embryos were fixed in 2% PFA and 2% glutaraldehyde in PBT buffer for 24 h and then embedded in plastic resin (JB-4 Plus) and sectioned at 5μm using a tungsten carbide knife on a rotary microtome. Sections were stained with Giemsa (1:20, Fisher Scientific) for 1 h followed by a 10 s acetic acid (0.05%) differentiation wash. Slides were dried and covered with Permount and imaged on a Nikon compound microscope fitted with a digital camera (VCU Biology microscopy core).

### 2.8. Micro Incisions in the Face

Prior to all surgeries, embryos were anesthetized in 0.1% tricaine (in 0.1× Modified Barth’s Saline (MBS)) for 15 min and then moved into clay-lined dissection dishes containing 0.01% tricaine. The embryos were placed vertically into custom-fit holes made in the clay and they were gently secured by modeling the clay around the face. Glass knives, created from pulled-out capillary tubes using a Sutter needle puller, were used to sever the tissue. Specifically, the pigmented epidermis and the fluffy white tissue below were cut by first inserting the knife (20–30 microns deep) at the edge of the stomodeum and cutting the tissues with an upward motion. This was repeated until the region surrounding the stomodeum was cut. Sham controls were performed by gently poking the embryo once with the same glass knife to the same depth and to the same region, but the tissues were not cut. To assess the effects on buccopharyngeal membrane perforation, embryos were maintained for 3–5 h (until complete rupture occurred in all control embryos) and imaged. To assess buccopharyngeal membrane structure, embryos were fixed in 4% PFA 30 min after surgery and labeled with phalloidin as described in Section 2.6. Phalloidin labeling, immunofluorescence auto fluorescent rendering.

## 3. Results

### 3.1. Buccopharyngeal Membrane Perforation in Xenopus Laevis

The embryonic mouth forms in a series of morphological steps, the last of which is the perforation of the buccopharyngeal membrane. The first indication of perforation occurred during stage 39 (~56–60 hpf at 23 °C) when a small hole was observed in the buccopharyngeal membrane (Figure 1A–C’). This first hole became larger and then a second hole often formed (Figure 1D,D’). As each of the holes increased in size, the cells appeared to stretch across the embryonic mouth (Figure 1E,E’). Eventually, the holes coalesced to form a single hole (Figure 1F,F’). Finally, this larger hole increased in size until the entire buccopharyngeal membrane disappeared. The entire process, from the first small hole to the elimination of the buccopharyngeal membrane, took approximately 5–6 h at 23 °C. 

### 3.2. Chemical Screen Identified Known and New Candidate Regulators of Buccopharyngeal Membrane Perforation

To uncover mechanisms regulating buccopharyngeal perforation and general mouth development, an unbiased chemical screening approach was employed. Embryos were exposed to 1200 different chemicals from the Cancer Diversity Set II and Natural Product Set I over the period of mouth development (Figure 1G). We determined that exposure to 41 different chemicals resulted in embryos with a persistent buccopharyngeal membrane but also an observable stomodeum (Appendix A), suggesting a more specific effect later in mouth development (Figure 1H–O, shows a subset). To find candidate targets of the identified chemicals, we turned to the biological activity database hosted in PubChem. From this, 176 different biological targets were identified, some chemicals having one target while others could target many proteins (such numerous targets present the possibility that the phenotypes observed are due to the perturbation of a combination of proteins). We then determined each target’s function and corresponding gene symbol by interrogating GeneCards (Appendix A). By manually scrutinizing the targets, several signaling molecules or transcription factors that are known to have important roles during embryonic development were identified (for example, members of the retinoic acid, BMP, WNT, and JAK/STAT signaling pathways, Figure 1P). Next, a functional classification of all the biological targets was performed using PANTHER, and categories were identified with five or more genes. Two such categories are known to be required for embryonic mouth development, WNT signaling and Apoptosis [2,14]. These results reflected the validity of the screen but additionally also allowed us to identify new candidate pathways to investigate. For example, enriched categories included opioid, dopamine, and serotonin signaling which could point to the importance of neurotransmission. Alternatively, ion channels and neurotransmitters have also been shown to have non-neural roles in craniofacial development [45,46]. Future work could further investigate how these neurotransmitters and other signaling pathways could be involved in buccopharyngeal membrane rupture.

### 3.3. Human Genetic Data from Patients with Choanal Atresia Identified Potential Regulators of Buccopharyngeal Membrane Perforation

In parallel, a complimentary but very different screening approach was utilized to uncover regulators of buccopharyngeal rupture with the hopes to narrow down the list from the chemical screen. Since the embryonic mouth develops in a similar way in mammals and amphibians, we hypothesized that the same factors regulate human and *Xenopus* buccopharyngeal membrane perforation. In humans, a persistent buccopharyngeal membrane on its own is rare and therefore there is no genetic data for this birth defect. There is genetic data for patients with Choanal Atresia, a condition that could be caused by a persistent buccopharyngeal membrane (Figure 2A). Thus, we used copy number variation data from patients with this birth defect to identify candidate regulators of buccopharyngeal membrane perforation. Using the DECIPHER database [43], 83 patients were identified with Choanal Atresia and had copy number deletions (Appendix A). Some of the deletions spanned large regions of chromosomes and included many different genes. Therefore, it is difficult to rule out the fact that many genes could contribute to Choanal Atresia in these patients. The total number of identified genes in this analysis was 2202 and these were compiled (Appendix A). First, we manually scrutinized the gene list for important developmental regulators. From this, we identified genes in the WNT, FGF, SHH, and JAK/STAT signaling pathways (Figure 2B). Notably, disruption of FGF signaling in particular has been reported in a connection between Choanal Atresia and Craniosynostosis [33]. In addition, in this DECIPHER screen, we also identified transcription factors known to be important in craniofacial development, such as LHX3, PAX8, and MSX1. A Panther functional classification of the affected genes was also undertaken as in Section 2. Strikingly, we noted that seven of the functional groups identified in the chemical screen were also identified in this DECIPHER screen (Figure 2C). Such common categories included the known pathways important in embryonic mouth development: Apoptosis and WNT signaling. In addition, Heterotrimeric G-protein signaling (Gi/Gs, Gq/Go), CCKR signaling, Huntington’s disease, and inflammation categories with identified in both screens. These results support the possibility that both screening approaches were effective in identifying known and new regulators of buccopharyngeal membrane rupture.

### 3.4. Overlap in Chemical and Genetic Screens Identifies JAK2 as a Candidate Regulator of Buccopharyngeal Membrane Perforation

In an attempt to focus our screening methods, the lists of genes identified in both approaches were narrowed down and compared. To select for genes that were most likely to be important in Choanal Atresia, we choose to focus on genes affected approximately 10% of the patients (Appendix A), which narrowed the list to 35 genes. In addition, to select for the most important targets in our chemical screen, we chose to focus on the proteins targeted by at least three different chemicals causing a persistent buccopharyngeal membrane. This narrowed this list to 28 different proteins. An overlap in these two smaller focused lists produced only one gene/protein in common, Janus Kinas 2 (JAK2) (Figure 2D). Therefore, we decided to test whether Jak2 could be a regulator of buccopharyngeal membrane perforation in *X. laevis*.

### 3.5. Deficient Jak2 Causes a Persistent Buccopharyngeal Membrane

To determine whether Jak2 is important for buccopharyngeal membrane perforation, modified antisense oligos (MO; Morpholino oligomer, Genetools) were first utilized to reduce this protein in developing embryos. The *jak2* MO targeted the transcriptional start sites of both *jak2* homeologs (*jak2.L* and *jak2.S*, Genetools, Appendix A). A range of *jak2* MO concentrations were injected into embryos at the one-cell stage and the faces were visualized at stages 40–41 after the buccopharyngeal membrane ruptured in control embryos. The morphant embryos had craniofacial defects that increased in severity with morpholino concentration. We determined that approximately 20 ng of MO/embryo resulted in 70% of embryos having a partially or fully intact buccopharyngeal membrane (not shown). At 40 ng of MO/embryo, 81.7% of embryos had either a fully or partially intact buccopharyngeal membrane (Figure 3A–A’’,B–B’’,F). Finally, at 60 ng/embryo, 93.3% of embryos had either a fully or partially intact buccopharyngeal membrane (Figure 3C–C’’,F). When one cell at the two-cell stage was injected with 30–40 ng of *jak2* MO, the buccopharyngeal membrane persisted on the injected side in 86.7% of these half morphants (Figure 3D–D’’,F) compared to 0% of the control half morphants. In addition, in these half-injected embryos, we noted a decrease in Jak2 protein in the injected half compared to the control half (see below). Thus, we provide the first evidence that Jak2 is required for buccopharyngeal membrane rupture.

### 3.6. Jak2 Is Specifically Required for Buccopharyngeal Membrane Rupture Rather Than Earlier Steps in Embryonic Mouth Development

One possible reason to explain why *jak2* morphants had a persistent buccopharyngeal membrane was that the MOs caused defects in earlier developmental events that non-specifically affected buccopharyngeal membrane rupture. To test for this possibility, we exposed embryos to a JAK2 inhibitor, Ruxolitinib, just prior to buccopharyngeal membrane perforation. Indeed, 81.7% of embryos exposed to this chemical had a partially or fully intact buccopharyngeal membrane, phenocopying the *jak2* morphants (Figure 3E–E’’,F). These results are also evidence that the MOs are specifically affecting Jak2 levels in the embryos. 

In addition, we also tested whether earlier events in embryonic mouth development were affected in *jak2* morphants by assessing the thickness of the buccopharyngeal membrane. During embryonic mouth development, the stomodeum becomes thinner to form the 1–2 cell layer thick buccopharyngeal membrane. Thus, it could be argued that earlier effects of Jak2 knockdown could affect buccopharyngeal membrane thinning which in turn could physically prevent its rupture. If this was true, then we expected to observe a thickened buccopharyngeal membrane in the *jak2* morphants. To view the buccopharyngeal membrane, embryos were rendered auto-fluorescent and transparent and then optical lateral sections were captured using confocal microscopy. Results indicated that the buccopharyngeal membrane was actually thin in 100% of the jak2 morphants analyzed with a persistent buccopharyngeal membrane (Figure 3G,H). These results suggest that decreased Jak2 did not affect earlier thinning of the stomodeum.

Together, the observations of buccopharyngeal membrane thickness and the late-stage JAK2 inhibitor treatments suggested that the reduction in Jak2 had a specific effect on the rupture of the buccopharyngeal membrane rather than on earlier events in embryonic mouth development.

### 3.7. Jak2 Is Required for Cranial Muscle Development

To uncover how Jak2 regulates buccopharyngeal membrane perforation, we first determined where Jak2 was acting during mouth development. We used a JAK2 antibody, created against the phosphorylated (tyrosine 1007/1008) human protein. Phosphorylation at this site is an indicator of the high activity of the JAK2 protein. Further, this region of the protein is 100% similar to the corresponding sequence of the *X. laevis* Jak2 protein (Appendix A). Robust labeling with the phosho-JAK2 antibody was observed in regions adjacent to the oral cavity just prior to and during buccopharyngeal membrane perforation (Figure 3I). This labeling was eliminated in *jak2* morphants (Appendix A). Such labeling appeared to be in the developing cranial muscles and indeed when we double-labeled, using a muscle-specific antibody, we observed overlap in phospho-JAK2 and muscle filaments (Figure 3I–L, arrows).

Since we observed active Jak2 in the cranial muscles of the developing embryos, we hypothesized that this protein is then required for cranial muscle formation. When the cranial muscles were examined in the *jak2* morphants, 96.7% of the embryos had a partial or complete reduction of muscles anterior to the eyes (Figure 3M–O). On the other hand, the trunk muscles did not appear affected in these morphant embryos, suggesting that Jak2 might have a specific role in cranial muscle development (Figure 3M,N). Certainly, cranial and trunk muscles are regulated in part by different developmental programs [47].

### 3.8. The Cranial Muscles Are Required for Buccopharyngeal Membrane Rupture

Our results thus far suggested that Jak2 was required for both cranial muscle development and buccopharyngeal membrane perforation. We, therefore, hypothesized that the development of the buccopharyngeal membrane rupture was dependent on the muscles surrounding the oral cavity. Supporting this hypothesis is the fact that the cranial muscles developed over the same time as buccopharyngeal membrane perforation (Figure 4A–C; [48]). At stage 38, just prior to buccopharyngeal membrane perforation, both the levator mandibulae longus (lml) and orbithyoidus (orb) muscles could be observed (Figure 4A). During perforation, these muscles increased in size and were joined by the angularis (an) muscles (Figure 4B). At the end of perforation, the cranial muscles were larger (Figure 4C), although the mouth did not yet appear to move.

To test whether cranial muscles are required for buccopharyngeal membrane perforation independent of Jak2, we performed experiments to perturb muscle development directly. To accomplish this, we first knocked down an important regulator of muscle development, MyoD. Embryos were injected with 40–50 ng/embryo of a previously validated *myod1* MO [44] at the one-cell stage. Results indicated that, indeed, 92% of *myod1* morphant embryos had a fully or partially intact buccopharyngeal membrane (Figure 4D,E–F’’).

To further demonstrate that muscles have a role in buccopharyngeal membrane perforation, we induced muscle paralysis. During this time, the jaws do not yet move, however, paralysis might perturb muscle tone and/or eliminate muscle twitching which could in turn affect muscle development. Embryos were exposed to *N*-Benzyl-p-toluenesulfonamide (BTS) which inhibits the ATPase activity of skeletal muscle myosin II and thereby reversibly blocks gliding motility and suppresses force and twitch production in fast skeletal muscle [49]. Embryos exposed to 100 um of BTS indeed resulted in paralysis. We found that this concentration was not lethal within 24 h since the effects could be reversed when embryos were removed from BTS (not shown). Results indicated that 58% of embryos treated with BTS from stage 37 until stage 40 had a fully or partially intact buccopharyngeal membrane (Figure 4D,G–H’’). The effects of BTS were not highly penetrant which could indicate that the contractibility of fast-twitch muscles itself is not the sole reason muscles are important for buccopharyngeal membrane rupture. It is also possible that the drug does not affect all individuals equally using a bath application.

One might also argue that the effects of the BTS on the buccopharyngeal membrane are secondary to its effect on heartbeat and circulation. To test whether a functional heart was required for buccopharyngeal membrane rupture, we removed the developing heart at stages 35–37. While this resulted in craniofacial differences, the buccopharyngeal membrane ruptured in 100% of the embryos (Appendix A). These results are further evidence that it is the developing cranial muscles that are important for buccopharyngeal membrane rupture during mouth development.

One possible reason for the importance of muscles is that they give the face its shape and size during craniofacial development. Thus, increasing cranial growth could in turn be structurally important for buccopharyngeal membrane perforation. If this is the case, then the buccopharyngeal membrane would not rupture in embryos with narrower and/or smaller faces (no matter the cause). To evaluate this possibility, we re-examined the images of embryos from the chemical screen (Figure 1, Section 2). There were several examples of embryos with narrower heads but also with a ruptured buccopharyngeal membrane (Appendix A). In addition, whether the buccopharyngeal membrane ruptured did not seem to depend on the size or shape of the mouth since even mouths that were small and round could have an absent buccopharyngeal membrane. These observations suggest that buccopharyngeal membrane rupture does not depend on the size or shape of the head and mouth, further supporting a more specific role of muscles in the process.

### 3.9. Jaw Muscles Are Connected to the Oral Epithelium by Laminin

To better understand how cranial muscles could influence buccopharyngeal membrane perforation, we next performed a more detailed examination of muscles with respect to the mouth and oral cavity. The muscles were labeled with the muscle-specific antibody (12/101) in transverse sections of the head during and just prior to and after buccopharyngeal membrane perforation. Further, these sections were counterstained with laminin which outlined the oral cavity and muscle compartments (Figure 5A–H). Intriguingly, in this analysis, we noticed that laminin connected the jaw muscle compartments to the basal lamina lining the oral cavity (Figure 5B,C,F,G, white arrows). Similarly, intimate associations between the oral cavity and muscles were also observed in histological sections made with 5 μm plastic sections and prepared with a general histological stain (Figure 5D,H, pink arrows). The epithelium of the oral cavity is continuous with the buccopharyngeal membrane. Therefore, these results suggest the possibility that jaw muscles might physically influence the oral cavity and thus the buccopharyngeal membrane.

Next, we performed a similar analysis of laminin in both the *jak2* and *myod* morphants. These results revealed the predicted absence or reduction in muscle fibers (reduction or lack of red labeling in Figure 5K,L). However, the laminin-lined compartments that would normally contain muscles were still present, although the laminin labeling appeared disjointed (Figure 5J–L, white arrows). Importantly, the compartments normally containing muscle were not joined to the oral cavity as in the controls. Such results might suggest that these connections might be integral to buccopharyngeal membrane rupture.

### 3.10. Incisions around the Oral Cavity Result in a Persistent Buccopharyngeal Membrane

To test whether the ECM connections between the muscle and oral cavity are indeed required for buccopharyngeal membrane rupture, we next attempted to sever such connections. Embryos were exposed to a low dose of tricaine in embryo media to decrease movement. Then embryos were inserted tail-first into holes made in clay-lined dishes. The clay was pinched around the head of each embryo to hold tightly. A fine capillary needle was used to make incisions around the edges of the embryonic mouth just prior to perforation (Figure 5M). The depth of the incisions was such that the connections between the mouth and muscle could be eliminated. Sham operations of embryos in the same clay-lined dish were performed, where the capillary knife only poked the epidermis leaving no observable cut in the epidermis. Following the surgeries, 50% of the media was exchanged to further reduce the concentration of tricaine and embryos were observed over 5 h. In 88% of the operated embryos, the buccopharyngeal membrane failed to perforate (Figure 5Q–S’). On the other hand, the buccopharyngeal membrane ruptured in all of the sham control embryos (Figure 5N–P’). Interestingly, approximately 30 min after the surgical incisions, the embryonic mouth seemed to collapse and become smaller (Figure 5R,R’). Therefore, in a subset of the experiments, some of the embryos were fixed 30 min post-surgery, sectioned, and labeled with fluorescently labeled phalloidin. In these labeled embryos, the buccopharyngeal membrane appeared to have folded over or buckled in embryos that had undergone the surgery. On the other hand, in the sham controls, the cells appeared as a normal linear structure covering the oral cavity (compare Figure 5T,U). These observations prompted the hypothesis that connections between the oral cavity and surrounding muscle generate force across the buccopharyngeal membrane. However, it should be noted that these experiments were not precise and could have resulted in the destruction of important signals, excess cell death, or general health of the embryo that led to a persistent buccopharyngeal membrane.

### 3.11. F-actin Puncta Were Observed in the Cells of the Buccopharyngeal Membrane Prior to Rupture

F-actin has been proposed to function as a dynamic tension sensor [50]. Therefore, if the cells of the buccopharyngeal membrane are under tension, then we hypothesized that we would observe changes in F-actin as the buccopharyngeal membrane perforates and ruptures. F-actin, labeled with fluorescently tagged phalloidin, lined the inner membrane of all epidermal cells and was enriched at the surface of ciliated cells (Figure 6A,B). At stage 37, the cells that will form the buccopharyngeal membrane appeared similar to the surrounding epidermis (Figure 6B,B’). However, at stage 39 just prior to perforation, we noted punctate accumulations of F-actin in cells of the buccopharyngeal membrane (Figure 6C,C’ white arrows). Later during perforation, F-actin puncta were observed (Figure 6D,D’, white arrows), as well as an enrichment around the perforating holes (Figure 6D,D’, white arrowhead). Thus, these results do indicate that there are changes in F-actin that could indicate that the buccopharyngeal membrane was under tension.

We next hypothesized that if the changes in the actin cytoskeleton of cells comprising the buccopharyngeal membrane are indeed due to tension across the structure, then embryos lacking cranial muscles would not display such changes. Thus, we examined F-actin in both *jak2* and *myod* morphants that have major reductions in jaw muscles. Results indicated that indeed 90% of the *jak2* and 80% of the *myod* morphant embryos appeared to have significantly less F-actin-positive puncta-like accumulations in the buccopharyngeal membrane compared to controls (Figure 6E–H’). In addition, we also observed more regions where actin appeared to be reduced between cells in the controls compared to the morphants (Figure 6F–H’). These results indicate that the presence of F-actin puncta and reduction in actin between cells of the buccopharyngeal membrane correlates with the presence of cranial muscles. However, we cannot exclude the possibility that actin polymerization occurs when cells are moving or changing shape as the buccopharyngeal membrane perforates. Such changes could be dependent on cranial muscles in some other way that is independent of mechanical forces.

### 3.12. Perturbing Actin Dynamics Results in a Persistent Buccopharyngeal Membrane

If the changes in the actin cytoskeleton were required for cells to sense or facilitate a response to tension, then we predicted that perturbing actin dynamics just before and during buccopharyngeal membrane perforation (stage 37–40) would prevent its rupture. First, actin polymerization was inhibited using cytochalasin D (1 μm) and we noted that 88.3% of embryos exposed to this compound had a fully or partially persistent buccopharyngeal membrane compared to 0% of the controls (Figure 6I,J’’,L). Interestingly, the mouth also appeared smaller, similar to the effects of the surgical incisions around the oral cavity (see Figure 5R). Muscle development was examined in a subset of embryos treated with cytochalasin D and no major observable defects were observed in muscle (Appendix A). Secondly, actin dynamics were also perturbed by inhibiting Rho-associated kinase (ROCK) which is critical to regulating actin organization [51]. We treated embryos with the ROCK inhibitor (ROCKOUT, 100 μM). Indeed, we determined that 91.7% of embryos exposed to this inhibitor resulted in a fully or partially persisting buccopharyngeal membrane (Figure 6K–K’,L). Notably, this treatment had dramatic effects on mouth development and resulted in changes in the mouth shape similar to what we observed in the *jak2* morphants (see Figure 3C). These experiments may provide evidence that actin dynamics are required for the rupture of the buccopharyngeal membrane. We cannot rule out that the effects we observed in response to perturbing actin and ROCK are actually due to effects on muscle development or function.

### 3.13. Apoptotic Cells Are Present in the Buccopharyngeal Membrane just Prior to Perforation

We next wondered how mechanical forces could cause perforation of the buccopharyngeal membrane. Previous work in the lab indicated that e-cadherin appeared to be reduced between some buccopharyngeal membrane cells and therefore could possibly create weak spots [37]. This reduction was similar to changes in F-actin (see Section 3.11. However, we also asked whether apoptosis played a role in weakening the connections between cells comprising the buccopharyngeal membrane. To answer this question, a marker of apoptosis, cleaved caspase-3, was examined. We labeled embryos at stages 38–39 before perforation began and observed that 9/20 embryos had one cleaved caspase-3-positive cell within the buccopharyngeal membrane (Figure 7A). These results indicate that the buccopharyngeal membrane does not undergo complete degeneration but rather a few dying cells may weaken some points within the structure.

Taken together, these results point toward a model (summarized in Figure 7B) where the cranial muscles, connected to the oral epithelium, are required to generate tension on the buccopharyngeal membrane. Weakened spots, caused by changes in adhesion and selective cell death, then permit the cells to be broken apart, allowing rupture to occur.

## 4. Discussion

Rupture of the buccopharyngeal membrane is a critical event in mouth development and establishes the first connection between the external environment and the inside of the embryo. Without this process, basic functions such as eating and breathing would not be possible in free-living embryos. In humans, failure of buccopharyngeal membrane rupture would disrupt the swallowing of amniotic fluid, which could also have multiple effects on embryonic development. Importantly, in all vertebrates, a lack of mouth opening could have additional complications in craniofacial development. For example, a persistent buccopharyngeal membrane has been proposed to lead to the formation of oral obstructions such as Choanal Atresia [1,31]. Therefore, understanding how the buccopharyngeal membrane disappears could lead to novel insights into such craniofacial birth defects. Moreover, by studying embryonic mouth development, we provide a deeper understanding of how craniofacial structures are integrated during development. This knowledge is not only critical for unraveling the complexity of craniofacial defects but also the function and evolution of the face.

We have uncovered that buccopharyngeal membrane rupture is intimately integrated with the formation of the cranial muscles. In particular, we demonstrated that disrupted jaw muscle development also perturbed buccopharyngeal membrane perforation. Muscles are also required for the development of other craniofacial structures such as cartilage and tendons [52,53]. Further support of a role for jaw muscle function in craniofacial development comes from the observations that humans and animal models with muscular dystrophy can also have a host of craniofacial malformations [54,55,56]. Thus, evidence is mounting to support a mechanical role for jaw muscles in various processes during craniofacial development [57].

How do the cranial muscles regulate buccopharyngeal membrane rupture? The first step to unraveling this question was the observation that the cranial muscles are physically connected to the oral cavity via the extracellular matrix (ECM). Specifically, laminin appeared to bridge jaw muscles with the oral epithelium that lines the oral cavity. Since this epithelium is continuous with the buccopharyngeal membrane, we hypothesized that the laminin connections allow for the muscles to influence this structure. To test this, we surgically severed the connections and determined that indeed they were important for buccopharyngeal membrane perforation. Importantly, when we did this, we also noticed that the mouth collapsed, and cells of the buccopharyngeal membrane buckled. Based on this data, we have formulated a model where the connections between the muscles and the oral epithelium create tension on the buccopharyngeal membrane and this is in turn necessary to stimulate its perforation. Such a hypothesis is supported by the fact that the tension sensor, F-actin [50], accumulated in the buccopharyngeal membrane cells during perforation. Moreover, this accumulation was reduced in embryos lacking cranial muscles and perturbation of actin dynamics caused a persistent buccopharyngeal membrane. Future experiments to test our hypothesis could include more targeted disruption of the ECM and measuring tension in the buccopharyngeal membrane directly. While no studies have specifically shown a connection between jaw muscles and buccopharyngeal membrane rupture in mammals, cranial muscles are in close association with the oral epithelium in mice [58]. Furthermore, in other amphibians, chickens, and humans, the cells of the buccopharyngeal membrane appear to be stretched across the mouth opening during its rupture [20,59,60]. Together, our work and that of others present the possibility that the buccopharyngeal membrane is under tension which is generated by jaw muscles.

How do the mechanical forces across the buccopharyngeal membrane stimulate its perforation? In other developmental events, mechanical forces can activate several different signals such as the Hippo pathway or Piezo-regulated calcium channels [61,62]. These in turn can initiate cytoskeletal changes, downregulate adherens junctions, and induce apoptosis [62,63,64,65,66,67]. We demonstrate here that there are indeed apoptotic cells in the buccopharyngeal membrane just prior to perforation. Dying cells have also been reported in the buccopharyngeal membrane of other vertebrates and have been postulated to generate weak spots necessary for perforation [3]. Consistently, apoptosis has been proposed to finetune other morphological processes in craniofacial development [68]. In addition to apoptosis, junctional remodeling between the cells comprising the buccopharyngeal membrane might also contribute to weakened regions. In previous work, we have also demonstrated that there are heterogenous reductions in junctional e-cadherin in the buccopharyngeal membrane prior to buccopharyngeal membrane perforation. This correlates with increased endocytosis and JNK signaling in the structure [1,2,37]. Therefore, we propose a model where decreased adhesion and programmed cell death create weak spots in the buccopharyngeal membrane that can then be pulled apart by the mechanical forces generated by the muscles (Figure 7).

In this study, we demonstrate the power of using human genetic data associated with birth defects combined with animal studies to uncover novel developmental mechanisms. We combined a chemical screen in *Xenopus* embryos with an analysis of human genetics associated with a birth defect that may be caused by a persistent buccopharyngeal membrane. This analysis uncovered the potential importance of JAK2 in mouth development. JAK2 is a Janus Kinase that acts as a cell surface receptor that can bind many different ligands including cytokines and growth factors and therefore can in turn regulate numerous processes in the embryo [69]. Mutations in Jak2 are associated with blood disorders and knockout mice are not viable due to the effects on hematopoiesis [70]. Because of this effect, many of the other developmental functions of JAK2 may be hidden or are not as well studied. Here, by using titratable tools in *Xenopus* embryos before blood has a significant role in craniofacial development, we have uncovered a novel role for Jak2. Specifically, JAK2 is required for the differentiation of jaw muscles. Consistently, JAK2 has been shown to regulate the differentiation of myoblasts which is partially mediated by muscle-specific transcription factors MyoD and MEF2 [71]. It will be interesting to determine how JAK2 is integrated into the muscle developmental program during craniofacial development.

## 5. Limitations of the Study

The embryonic or primary mouth is a difficult structure to study in all vertebrates including *Xenopus.* Its location is especially challenging and, to view this structure, the embryo needs to be positioned in a substrate to position the face in a manner amenable to microscopy. Since the mouth forms during a tadpole swimming stage, anesthesia is also required. This is problematic since tricaine may alter muscle function and preliminary experiments suggest it does disrupt BPM rupture. Perforation of the buccopharyngeal membrane also takes several hours and occurs as the craniofacial prominences are growing, forcing the buccopharyngeal membrane to greater depths in the head. For all of these reasons, imaging and access to the buccopharyngeal membrane in the live embryo are not amenable. Therefore, methods to measure tissue stiffness and forces directly using FRET sensors or atomic force microscopy have proven technically difficult. Thus, our hypothesis is largely based on correlative and observational data. We hope that this work provides the basis for more effort and collaboration to establish direct methods to test our model. Both the genetic and chemical screens also have broad limitations in that malformations in mouth development may be due to multiple effects. For example, in the chemical screen, each chemical could affect a number of targets that together are responsible for a persistent buccopharyngeal membrane. In the genetic screen, a single copy number variation could affect numerous genes and in combination may result in Choanal Atresia in patients. More extensive validation of these screens is necessary to resolve such limitations.

## 6. Conclusions

This work uncovers a novel role for Jak2 in regulating cranial muscle development, which in turn is required for the final step in embryonic mouth development, buccopharyngeal membrane perforation. We hypothesize that the jaw muscles generate tension across the buccopharyngeal membrane. This tension or force, together with weakened spots due to apoptosis and junctional remodeling, pulls the cells apart to create the mouth opening (see graphical abstract).

## 7. Patents

There is no patent resulting from the work reported in this manuscript.

## Figures and Tables

**Figure 1 jdb-11-00024-f001:**
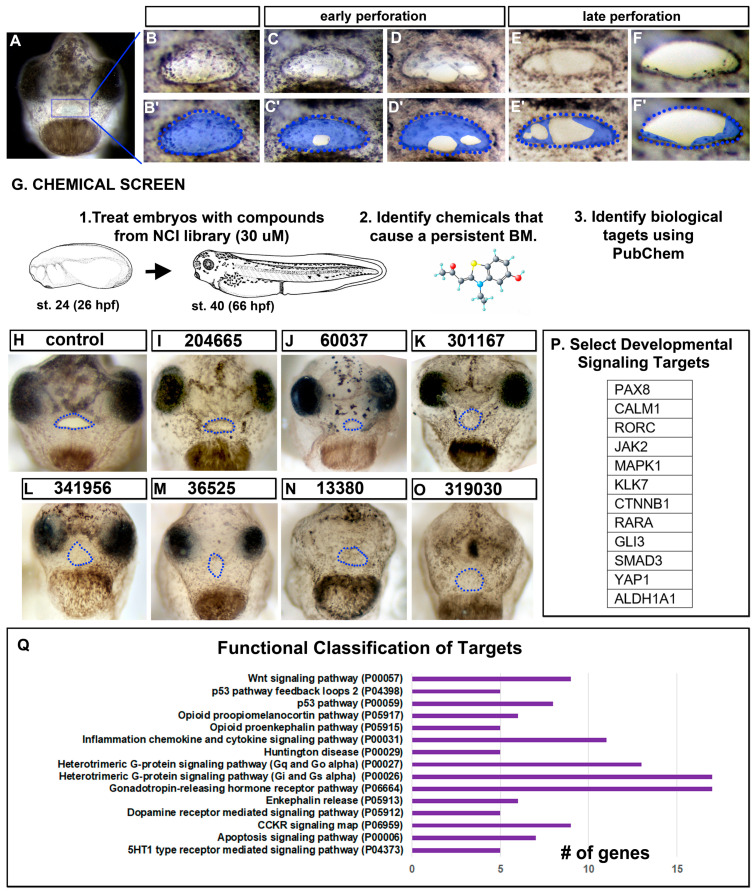
Buccopharyngeal membrane rupture and chemical screen. (**A**) Frontal view of the face of an embryo at stage 39 prior to the perforation of the buccopharyngeal membrane. (**B**–**F’**) magnified images of the mouth at progressive points of buccopharyngeal membrane perforation. In the prime labeled images, the buccopharyngeal membrane is shaded blue. Each image is from a different embryo that represents the most common appearance at each stage (from over 200 embryos examined in 10 biological replicates). (**G**) Schematic outlining the chemical screen. (**H**–**O**) A subset of representative embryos treated with select chemicals causing a persistent buccopharyngeal membrane but with a stomodeum present. The presumptive mouth is outlined in blue dots. The National Service Center Number identifier is shown above each embryo. (**P**) Protein symbols of select targets of chemicals causing a persistent buccopharyngeal membrane. (**Q**) The top functional categories identified from the targets of chemicals causing a persistent buccopharyngeal membrane. The GO identifier is in brackets. Abbreviations: # = number, st. = stage.

**Figure 2 jdb-11-00024-f002:**
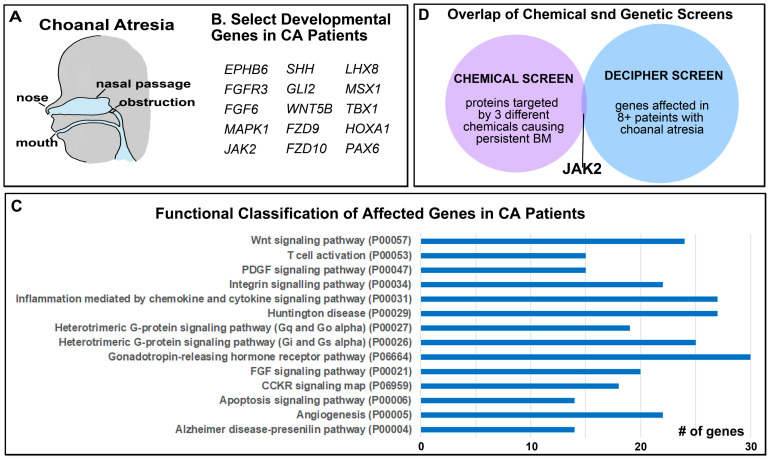
Copy Number Variants in patients with Choanal Atresia. (**A**) Schematic showing where the airway blockage can occur in Choanal Atresia. (**B**) Select genes with copy number deletions identified in patients with Choanal Atresia known to be important in development. (**C**) The top functional categories identified from copy number deletions in patients with Choanal Atresia. The GO identifier is in brackets. (**D**) Overlap in genes identified in the chemical and genetic screens identifies JAK2. Abbreviations: # = number.

**Figure 3 jdb-11-00024-f003:**
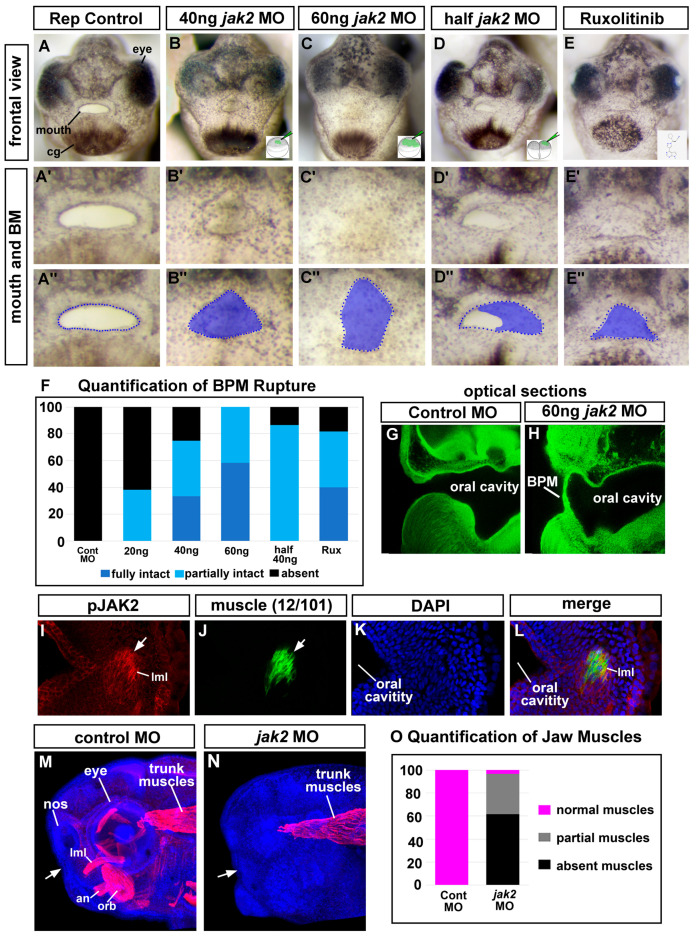
Jak2 knockdown affects the buccopharyngeal membrane and cranial muscles. (**A**–**E’’**) Frontal view of the face of representative embryos at stages 40–41 showing ruptured or persisting buccopharyngeal membranes (images from *n* = 60, 3 biological replicates for each treatment). The bottom right corner shows an image of the injection site/stage. Prime-labeled images show magnified images of the mouth and the double-prime-labeled images show the same images with the buccopharyngeal membrane shaded in blue. (**F**) Shows relative percentages of embryos with an intact, partially intact, or absent buccopharyngeal membrane (*n* = 60 for each group, 3 biological replicates). (**G**,**H**) Optical sagittal section through the head of representative controls and *jak2* morphants. The buccopharyngeal membrane is absent in the control and is present but thin in the *jak2* morphants (*n* = 12, 2 biological replicates). (**I**–**L**) Representative thick agarose section through the face showing phosphor-‘JAK2 and 12/101 immunofluorescence. (**I**) phospho-Jak2 (red), (**J**) 12/101 = muscle-specific antibody (green), (**K**) DAPI (blue), (**L**) merge. (**M**,**N**) Lateral views of representative embryos showing 12/101 muscle labeling (red) in control and *jak2* morphants and counterstained with DAPI (blue). White arrows point to the location of the mouth. (**O**) Quantification of the presence of cranial muscles in control and *jak2* morphants (*n* = 60, 2 biological reps). Abbreviations; BPM = buccopharyngeal membrane, nos = nostril, MO = morpholino, levator mandibulae longus = lml, orbithyoidus = orb, angularis = an.

**Figure 4 jdb-11-00024-f004:**
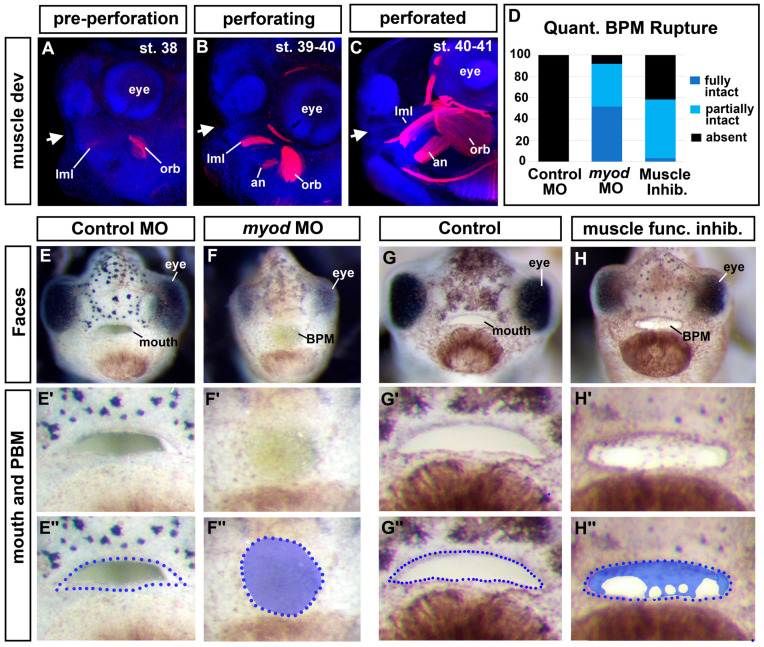
Cranial muscles are required for buccopharyngeal membrane rupture. (**A**–**C**) Lateral views of embryos showing cranial muscles (red) and counterstained with DAPI (blue) (best images from 10 embryos imaged at each stage). White arrows show the location of the developing mouth. (**D**) Proportion of embryos that had a fully intact, partially intact, or absent buccopharyngeal membrane after myoD knockdown or treatment with a muscle inhibitor (BTS) compared to controls (*n* = 60, 3 biological replicates for each treatment). (**E**–**H’’**) Frontal views of representative embryos injected with myoD morpholinos or treated with an inhibitor of muscle function (BTS). Prime-labeled images show magnified images of the mouth, and the double-prime-labeled images show the buccopharyngeal membrane shaded in blue. Abbreviations: levator mandibulae longus = lml, orbithyoidus = orb, angularis = an, BPM = buccopharyngeal membrane.

**Figure 5 jdb-11-00024-f005:**
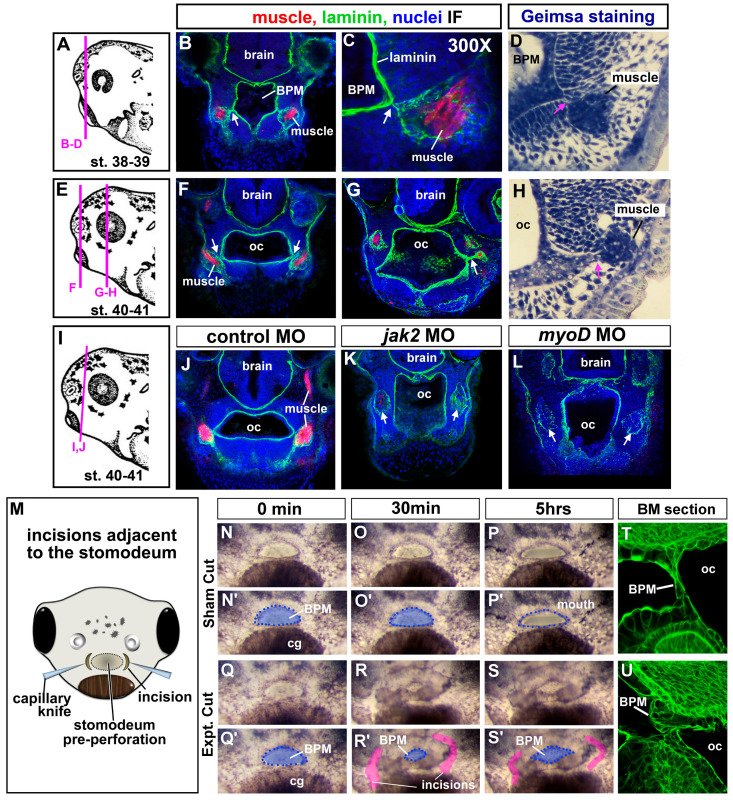
(**A**–**L**) Muscle and oral cavity connections. (**A**,**E**,**I**) Schematics of the lateral views of embryos showing locations of the corresponding sections. (**B**,**C**,**F**,**G**,**J**–**L**) Transverse sections labeled with antibodies to detect muscle-specific protein (12/101 in red), laminin (green), and counterstained with DAPI (blue). Images were chosen from representative images taken from 20 different embryos in 2 biological replicates. White arrows indicate laminin that bridges the oral cavity and muscle compartments. (**D**,**H**) Plastic section stained with Giemsa showing representative images of the association between muscle and oral cavities (based on sections of 20 embryos in 2 biological replicates). Pink arrows indicate the connections between the muscle and the oral cavity. (**M**) Schematic of a frontal view showing the location of the surgical incisions. (**N**–**S’**) Shows the embryonic mouth and buccopharyngeal membrane in representative embryos before (0 min) and after sham or incisions (at 30 min and 5 h). Prime images show the buccopharyngeal membrane shaded in blue and the mouth outlined in blue dots. Representative images chosen from 50 embryos performed over 5 biological replicates. In R and S, the incised tissue is colored pink. (**T**,**U**) representative embryos 30 min after surgeries (or sham) sectioned and labeled with phalloidin to show the cellular arrangements in the buccopharyngeal membrane. Shows representative images taken from a total of 10 embryos in 2 biological replicates, Abbreviations: oc = oral cavity, BPM = buccopharyngeal membrane, cg = cement gland.

**Figure 6 jdb-11-00024-f006:**
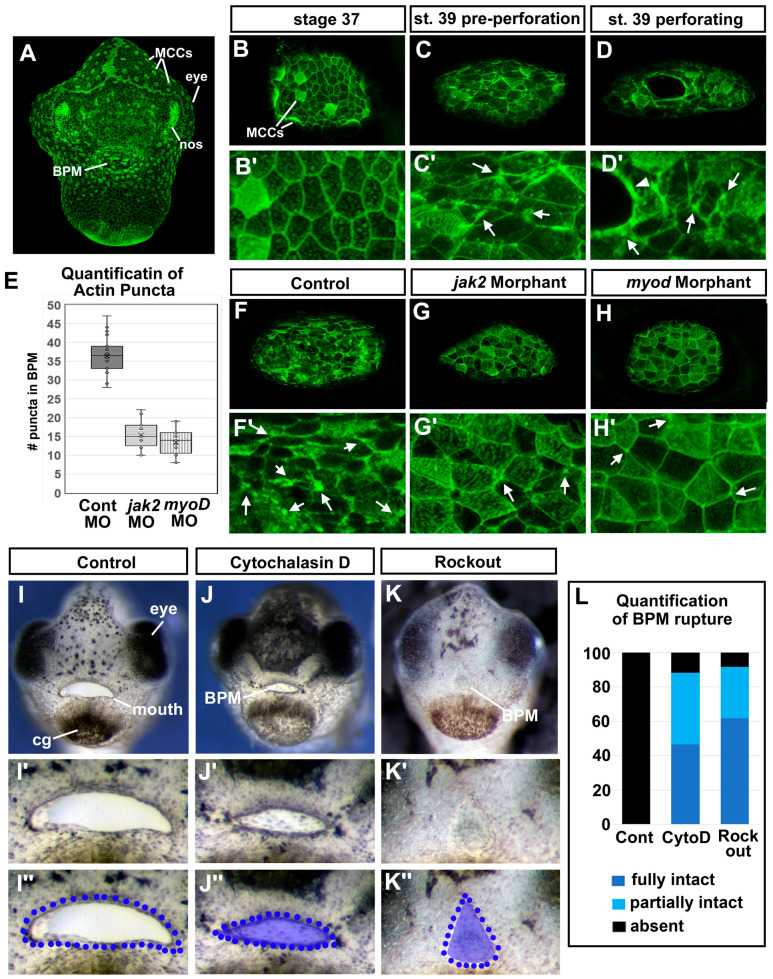
Actin dynamics and buccopharyngeal perforation. (**A**–**G**) Actin dynamics revealed by phalloidin labeling during buccopharyngeal membrane perforation. (**A**) Shows a frontal view of the face at stage 39 (representative from 30 embryos in 4 biological replicates). (**B**) The buccopharyngeal membrane at stage 37 when the mouth shape is round. (**C**) The buccopharyngeal membrane at stage 39 just prior to perforation. Note the puncta spots and accumulation of F-actin (white arrows). (**D**) The buccopharyngeal membrane during perforation shows accumulation of F-actin surrounding the holes. (**E**–**H’**) The buccopharyngeal membranes of control morphants (**F**), *jak2* morphants (**G**), and *myod* morphants (**H**). (**I**–**K’’**) Images of representative embryos after treatment with cytochalasin D (1 μM) and Rockout (100 μM) from stages 37–40. Quantification of buccopharyngeal membrane rupture in embryos exposed to actin inhibitors shown in (**L**) (*n* = 60, 3 biological replicates).

**Figure 7 jdb-11-00024-f007:**
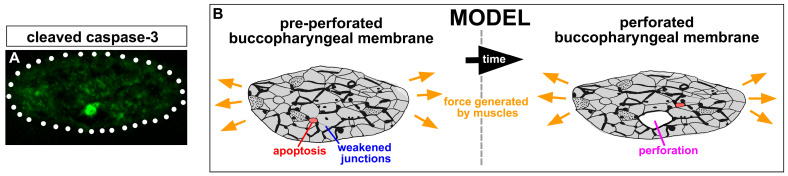
Apoptosis and Model: (**A**) Cleaved caspase-3-positive cell (green) in the buccopharyngeal membrane pre-perforation (representative of 9 embryos, *n* = 20, 2 biological replicates). (**B**) Forces from the muscles generate tension on the buccopharyngeal membrane. Weakened spots, caused by weakened junctions and selective apoptosis, then permit the cells to be broken apart, allowing perforation.

## Data Availability

Large data sets are available here as part of the Appendix A. All other data not included in the manuscript is available upon request.

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
