# Peer review of "Jak2 and Jaw Muscles Are Required for Buccopharyngeal Membrane Perforation during Mouth Development"

_jdb, 2023, doi:10.3390/jdb11020024_

Round 1

Reviewer 1 Report

In this study, the authors found that Jak2 is required for mouth formation in Xenopus. They screened for chemicals that cause perforation of the buccopharyngeal membrane and also screened for genes associated with cleft palate in the DECIPHER database. By combining these two different screens, we found that Jak2 is a regulator of perforation. In fact, injection of jak2MO was shown to inhibit perforation and cranial muscle formation. To demonstrate a direct relationship between jaw muscles and buccopharyngeal membrane perforation, the authors observed muscle disruption by myoD MO, muscle paralysis by inhibitor treatment, and inhibition of perforation formation by manual microincision. In addition, the distribution of F-actin was examined and the relationship between perforation and mechanosensing was discussed.

Since this study was well designed and many experiments were performed to obtain each result, the conclusions are essentially reliable. After the author has properly addressed the reviewers' comments, this study is worthy of publication in JDB.

Major comments:

L238/Fig. 1A-F: If the muscles on both sides of the BPM stretched the BPM directly (via laminin, of course), it seems unlikely that the starting point of the rupture would be the center of the BPM (e.g., the edge of the BPM), unless there is heterogeneity in the stiffness of the BPM.

L238/Fig. 1H-O: Phenotypes of chemically treated tadpoles appear to vary. For example, tadpoles treated with 204665/60037 have no perforation but BPM is observed, whereas tadpoles treated with 36525/13380 have complete loss of BPM. Do these phenotypic differences reflect only severity or functional mechanisms? Classification of these phenotypes may provide insight into the mechanism of rupture.  

L341/Fig. 3M: The author should indicate the location of the muscles focused on in this study.

L483/Fig.5Q-S: The method for sham cut should be shown in more detail (were other areas cut? or different experiments?).

L508/Fig.6C and E-G: It is difficult to tell which cells are showing "punctate accumulation". They need to be indicated by arrows. Also, the author's claim that there is a difference between E and F/G is not convincing.

L511-512: It is difficult to conclude from the phalloidin staining results alone that there is tension in the BPM, as there may be other reasons (e.g., simply a change in degree of polymerization).

L543/Figure 6I: Cytochalasin administration may affect jaw muscle formation. 12/101 staining of cytochalasin-treated tadpoles should be performed.

Minor:

L457: White arrow not visible in Figure 5J-L.

L496: R and S->R' and S'

Reviewer 2 Report

The manuscript entitled "Jak2 and jaw muscles are required for mouth development" focuses on the role of Janus kinase 2 (Jak2) and jaw muscles in mouth development, specifically in the process of buccopharyngeal membrane rupture. The author hypothesized that muscles are required to exert tension across the buccopharyngeal membrane, and such tension is necessary for its perforation.

There are some minor comments:

Line 105. Add MBS abbreviation.

Line 115-116. More information about "the NCI Cancer Diversity set II" is needed. For example, to present who the supplier/manufacturer, a link to a full description of this set. What was the final concentration of DMSO in the wells?

Line 121 Replace “as described below” with “as described in Section 2.4. Imaging the Facial Features”.

Line 131, 138. The links provided do not work. It is recommended to use a more reliable provider to store scientific data rather than a personal google drive.

Line 199. Replace “as described above” with “as described in Section 2.6. Phalloidin labeling, immunofluorescence auto fluorescent rendering”

Line 216.Cancer Diversity Set II and Natural Product Set I”. Please provide a link about it. Unfortunately, it was not possible to determine from the name which substances were used.

Figure 1. The PubChem logo must be replaced.

Line 246. The National Service Center Number identifier. Please provide a link about it. It is not clear what kind of chemistry this is, or what National Service Center Number the article is referring to.

Line 300. Add to “MO” “(Morpholino oligomer)”.

Line 313. Replace “(see below)”.

Reviewer 3 Report

I would like to appreciate author for conducting this study. I hereby given few suggestions to make the article scientifically sound and easy to understand.

1. The title of the article looks very general. I would like the author to consider modification of the study title.

2.I would like to suggest the author to find reference articles that are less than 10 years old 

31/60 reference articles were older than 10 years 

3.In introduction section , it would be helpful for the readers if the author provide some information on the novelty of this study in a brief manner.

4.I strongly suggest the author to provide ethical considerations of this study

5. Please provide information on the other models that help to provide better information on mouth development in table format. [ Techniques, advantages and disadvantages ]

6.Please provide the limitations of this study.

Moderate English editing needed .

Round 2

Reviewer 1 Report

Present version of the manuscript is appropriately revised, so there are nothing more to be pointed out.